# Evaluation of Cell-Free Synthesized Human Channel Proteins for In Vitro Channel Research

**DOI:** 10.3390/membranes13010048

**Published:** 2022-12-30

**Authors:** Rei Nishiguchi, Toyohisa Tanaka, Jun Hayashida, Tomoya Nakagita, Wei Zhou, Hiroyuki Takeda

**Affiliations:** 1Proteo-Science Center, Ehime University, Bunkyocho 3, Matsuyama 790-8577, Ehime, Japan; 2Nissan Chemical Corporation, Shiraoka 1470, Shiraoka 349-0294, Saitama, Japan

**Keywords:** cell-free membrane protein synthesis, proteoliposome, voltage-gated potassium channels, planar lipid bilayer assay, heteromeric complex, protein array

## Abstract

Despite channel proteins being important drug targets, studies on channel proteins remain limited, as the proteins are difficult to express and require correct complex formation within membranes. Although several in vitro synthesized recombinant channels have been reported, considering the vast diversity of the structures and functions of channel proteins, it remains unclear which classes of channels cell-free synthesis can be applied to. In this study, we synthesized 250 clones of human channels, including ion channel pore-forming subunits, gap junction proteins, porins, and regulatory subunits, using a wheat cell-free membrane protein production system, and evaluated their synthetic efficiency and function. Western blotting confirmed that 95% of the channels were successfully synthesized, including very large channels with molecular weights of over 200 kDa. A subset of 47 voltage-gated potassium ion channels was further analyzed using a planar lipid bilayer assay, out of which 80% displayed a voltage-dependent opening in the assay. We co-synthesized KCNB1 and KCNS3, a known heteromeric complex pair, and demonstrated that these channels interact on a liposome. These results indicate that cell-free protein synthesis provides a promising solution for channel studies to overcome the bottleneck of in vitro protein production.

## 1. Introduction

Channel proteins are important membrane proteins that regulate ion and water permeability and control voltage potential across cell membranes in response to stimuli such as voltage changes, ligand binding, temperature, and pressure [1,2,3]. Channels play essential roles in various organs, including neurological systems, cardiovascular systems, musculoskeletal systems, gastrointestinal and metabolic systems, and respiratory systems. Therefore, they are considered important therapeutic targets for a variety of diseases, such as arrhythmias, hypertension, pain, stroke, CNS diseases, autoimmune diseases, and diabetes [1,3,4]. Channels are the second largest drug target family, accounting for about 18% of all human protein drug targets [5]. Additionally, channels are important in drug safety assessments. For example, many drugs with cardiac-related side effects have been shown to block hERG ion channels, making studies on hERGs important for lead optimization in early drug discovery studies [6]. Thus, technologies for analyzing the function of channels and evaluating drugs that act on channels are essential for drug discovery and development.

Sample preparation is one of the obstacles to in vitro channel research. Channels are membrane proteins with 2 to 24 transmembrane helices, and many of them form dimeric or tetrameric complex [1]. Because of their highly complex structures, they are difficult to express and often unstable in cellular expression systems. Moreover, because of their roles in cellular signaling and related regulatory pathways, the overexpression of channels interferes with cell functions, often leading to abnormalities in homeostasis.

Cell-free protein synthesis, a technology that comprises transcription and translation reactions in vitro, has attracted attention as a promising tool to overcome the difficulties in channel research. Cell-free systems offer several approaches for membrane protein synthesis. The open reaction system allows the use of additives such as hydrophobic scaffolds that support the proper folding of membrane proteins. In the last decade, the in vitro translations of various membrane proteins have been reported [7,8,9,10,11]. We have also developed a membrane protein synthesis method using liposomes added to a wheat cell-free system [12,13]. Several groups have attempted the cell-free synthesis of channels and have indeed reported the successful synthesis of functional channels [14,15,16,17,18,19]. However, considering the diversity of the channel structures and functions, reported cases are still insufficient for determining which channel protein proteins classes to which cell-free synthesis can be applied.

In this study, we collected 250 human channels and synthesized them using a wheat cell-free membrane protein synthesis system. In addition, we evaluated potassium ion channels in a planar lipid bilayer (PLB) assay, an electrophysiological technique to monitor the behavior of ion channels at the single-molecule level in vitro [20,21]. Furthermore, we investigated the formation of heteromeric channel complexes on liposomes. Based on these results, we discuss the promising applications and limitations of the cell-free synthesis of channels for channel research.

## 2. Materials and Methods

### 2.1. Production of 8 Voltage-Gated Potassium Ion Channels by Bilayer Dialysis Method

Five K_V_ channels and three K_2P_ channels were synthesized using a wheat cell-free membrane production system following the bilayer dialysis method, as described previously [12,13]. The cDNA of KCNA1, KCNA2, KCNA5, and KCNK5 were sourced from Mammalian Gene Collection [22]. The cDNA of KCND2, KCNK2 and KCNK4 were prepared by gene synthesis (Thermo Fisher Scientific, Waltham, MA, USA). These cDNAs were inserted into the expression vectors of pEU-E01-MCS and pEU-AGIA-GW for the wheat cell-free system using Gibson Assembly reagent (New England Biolabs, Ipswich, MA, USA) [23]. Plasmids were prepared using the NucleoBond Xtra Midi kit (Takara Bio, Kusatsu, Japan), phenol/chloroform purified, ethanol precipitated, and then dissolved in ultrapure water to 1 mg/mL. A WEPRO7240 Expression Kit (CellFree Sciences Co., Ltd., Yokohama, Japan) was used to perform the transcription and translation reactions. Lyophilized asolectin liposomes are also commercially available from CellFree Sciences. In brief, the transcription reaction mixture was prepared by mixing 86.4 µL of ultrapure water, 30 µL of transcription buffer LM, 15 µL of NTP mixture (25 mM each), 1.8 µL RNase Inhibitor, 1.8 µL SP6 polymerase and 15 µL of the expression vector (1 mg/mL), which was then incubated at 37 °C for 6 h. The translation reaction mixture was prepared by mixing 130 µL of SUB-AMIX SGC translation buffer, 20 µL of creatine kinase (2 mg/mL, Sigma-Aldrich), 100 µL of asolectin liposome (50 mg/mL), 125 µL of WEPRO7240 wheat germ extract, and 125 µL of the transcription product. A dialysis cup (10k MWCO Slide-A-Lyzer MINI Dialysis Devices, 2 mL, Thermo Fischer Scientific) was inserted into 3 mL of translation buffer. Two mL of translation buffer and 500 µL of the translation reaction mixture were layered in the dialysis cup. The reaction was incubated at 22 °C for 20 h.

After the translation reaction, the suspension in the dialysis cup was transferred to a 1.5 mL tube and centrifuged at 17,800× *g* and 4 °C for 15 min. The proteoliposome pellet was washed three times with an HBS buffer (20 mM Hepes-NaOH, 150 mM NaCl, pH 7.2) before being resuspended in 400 µL of HBS buffer.

### 2.2. Density Gradient Centrifugation

The integration of the ion channels into the liposomes was evaluated by using iohexol density gradient centrifugation. Three hundred µL of the purified proteoliposome suspension prepared in Section 2.1 was mixed with the same volume of 80% iohexol (TCI, Tokyo, Japan)/PBS solution to obtain a final concentration of 40% iohexol. As shown in Figure 1A, 600 μL of 40% iohexol/proteoliposome/PBS, 650 μL of 35% iohexol/PBS, 650 μL of 30% iohexol/PBS, and 100 μL of PBS were layered in a 2 mL ultracentrifuge tube. Ultracentrifugation was performed at 206,000× *g* and 20 °C for 1 h. Subsequently, two hundred µL of the solution were collected from the top and bottom of the tube, respectively. The samples taken from the crude translation reaction mixture, centrifugation-purified proteoliposomes, and the top and bottom fractions of iohexol density gradient centrifugation were subjected to SDS-PAGE analysis. SDS-PAGE gels were stained using Coomassie Brilliant Blue G-250 (Nakarai tesque, Kyoto, Japan).

### 2.3. Single Channel Current Recording by PLB Assay

All of the PLB assays were performed using an Orbit mini device (Nanion Technologies GmbH., Munich, Germany) and a MECA 4 Recoding Chip, 100 µm (Ionera, Freiburg, Germany). A potassium-based buffer (150 mM KCl, and 20 mM Tris-HCl (pH7.5)) was injected into the chamber of the MECA 4 chip, and the air was removed from each microcavity by gently pipetting. A planar lipid bilayer was created on each microelectrode cavity by coating them with 1,2-Diphytanoyl-sn-Glycero-3-Phosphatidylcholine (DPhPC, Avanti Pollar Lipids Inc., Alabaster, AL, USA) or asolectin (Sigma-Aldrich, St. Louis, MO, USA) solubilized in decan at a 10 µg/mL concentration. During this step, the condition of the membrane was monitored using Orbit mini with elements data reader software. Finally, 5µ L of purified proteoliposome was injected into the chamber while recording the signal variation of the voltage for ten minutes to one hour.

All of the PLB data were analyzed using Clampfit 10.4 software (Molecular Devices, Palo Alto, CA, USA). First, the obtained signals were compensated for by making the baseline 0 pA and 0 mV, then all points of the signals were analyzed by plotting histograms. For the I-V plot, the average amplitude was calculated by fitting the Gaussian function against all points of the histograms.

### 2.4. Cell-Free Synthesis of 250 Human Channels by Bilayer Method

The cDNA clones coding human channels were collected from cDNA resources at the Kazusa DNA Research Institute [24] (222 clones) and the Mammalian Gene Collection [22] (28 clones). The cDNAs were subcloned into vector pEU-E01-FLAG. After subcloning, the *E. coli* clones transformed by each expression plasmid were arranged on 96-well plates and stored as glycerol stocks. All of the subsequent steps, such as PCR, transcription, and translation, were performed in the 96-well format using the Liquidator 96 pipetting apparatus (Mettler Toledo, Columbus, OH, USA). To prepare the transcription templates, the DNA fragments were amplified by PCR using PrimeStar Max PCR polymerase (Takara Bio), primer SPu-2 (5′-CAGTAAGCCAGATGCTACAC-3′), along with primer AODA2306 (5′-AGCGTCAGACCCCGTAGAAA-3′) and with glycerol stocks diluted 1/40 with TE buffer as a template. The transcription reaction mixtures were prepared by mixing 6.16 µL of ultrapure water, 2.8 µL of transcription buffer LM, 1.4 µL of NTP mixture (25 mM each), 0.28 µL RNase Inhibitor, 0.56 µL SP6 polymerase, and 2.8 µL of the PCR product in a 96-well PCR plate. The transcription reaction was incubated at 37 °C for 18 h. The translation was conducted using the bilayer method with some modification [25]. The translation reaction mixtures were prepared by mixing 12.5 µL of the transcription product, 8 µL of WEPRO 7240 wheat germ extract, 0.1 µL of creatine kinase (20 mg/mL), 0.5 µL RNase Inhibitor, and 5 µL of 50 mg/mL asolectin liposome. Twenty-five µL of the translation reaction mixture and 125 µL of SUB-AMIX SGC translation buffer were layered in 96-well plates. The translation reactions were incubated at 22 °C for 20 h. After the completion of the translation reactions, the reaction mixtures were split into portions and then frozen in liquid nitrogen and stored at −80 °C. The expression of each protein was confirmed by Western blotting using an HRP-conjugated anti-DYKDDDDK tag antibody (FUJIFILM Wako Pure Chemical, Osaka, Japan).

### 2.5. In Vitro Co-Synthesis and Immunoprecipitation of KCNB1 and KCNS3

The AGIA tag sequence (EEAAGIARPL) [26] was fused to the N-terminal of FLAG-KCNB1 by inverse PCR using pEU-E01-FLAG-KCNB1 as a template. The AGIA-FLAG-KCNB1 and FLAG-KCNS3 co-synthesis was performed as described below. The translation reaction mixture was prepared by mixing 17.8 µL of the translation buffer, 1.2 µL of 2 mg/mL creatine kinase, 12 µL of 50 mg/mL asolectin liposome, 15 µL of WEPRO7240 wheat germ extract, and 15 µL of the transcription reaction. For co-synthesis, equal amounts of AGIA-FLAG-KCNB1 and FLAG-KCNS3 transcription reaction were mixed and added to the translation reaction. A dialysis cup (10 k MWCO Slide-A-Lyzer MINI Dialysis Devices, 100 µL, Thermo Fischer Scientific) was inserted into 1 mL of the translation buffer SUB-AMIX-SGC. A total of 300 µL of translation buffer and 60 µL of the translation reaction mixture were layered in the dialysis cup. The reaction was incubated at 22 °C for 20 h. Proteoliposomes were washed three times with HBS buffer before being resuspended in 125 µL of HBS buffer. To solubilize the proteoliposomes, 125 µL of a solution containing 20 mM Hepes-NaOH, pH 7.2, 150 mM NaCl, 1% *n*-dodecyl-β-D-maltoside (Dojindo, Kumamoto, Japan), 0.2% cholesteryl hemisuccinate (Sigma-Aldrich), 10% glycerol (Nakarai tesque), 1 mM DTT, and 2.5 µL of protease-inhibitor cocktail (Nakarai tesque) were added to the proteoliposomes suspension. The mixture was rotated gently at 4 °C for 1 h, then centrifuged at 17,800× *g* and 4 °C for 15 min. The supernatant was collected into a new tube and applied to immunoprecipitation. Twenty µL of Dynabeads Protein G (Thermo Fisher Scientific) were washed three times with 200 µL of washing buffer (20 mM Hepes-NaOH, pH 7.2, 150 mM NaCl, 0.1% *n*-dodecyl-β-D-maltoside). Four µg of Anti-AGIA monoclonal antibody [26] were added to the beads and incubated at 4 °C for 1 h while applying gentle rotation. The beads were washed three times with 200 µL of washing buffer. Then, 150 µL of the solubilized ion channel micelle and 350 µL of HBS buffer were added to the beads and incubated at 4 °C for 1 h while gently rotating the mixture. The beads were washed three times with 200 µL of HBS buffer, then resuspended in 100 µL of HBS buffer. The beads suspension was transferred to a new tube, mixed with 150 µL of SDS-PAGE sample buffer, and incubated at 70 °C for 5 min. Input and IP samples were applied to Western blotting using HRP-conjugated anti-DYKDDDDK tag antibody.

## 3. Results

### 3.1. Cell-Free Synthesis of K_V_ and K_2P_ Potassium Ion Channels

First, we synthesized several voltage-dependent potassium ion channels in our cell-free membrane protein production system to examine whether the channels were synthesized well and integrated into the liposomes. The integration of membrane proteins with the appropriate hydrophobic scaffolds is critical for the expression of the activity of membrane proteins [7,27]. Voltage-gated potassium ion channels (K_V_ channels) and two-pore-domain potassium ion channels (K_2P_ channels) are well-studied ion channels [28,29,30,31,32]. The K_V_ channel consists of tetrameric pore-forming α-subunits and regulatory β-subunits, where the α-subunit is a membrane protein with six transmembrane helices. The pore-forming α-subunit of the K_V_ channels has a pore-forming loop and six transmembrane helices, and four α-subunits form a functional pore in a tetramer architecture [28]. The K_2P_ family has two pore-forming loops and four transmembrane helices; two subunits form a dimer.

We synthesized five K_V_ channel α-subunits, KCNA1, KCNA2, KCNA5, KCND2, and KCNH2, and three K_2P_ channels, KCNK2, KCNK4, and KCNK5, in the wheat cell-free system using the bilayer dialysis method (Figure 1A). After translation, the proteoliposomes were purified by using centrifugation and a buffer rinse, then we confirmed the synthesized channels by SDS-PAGE and Coomassie Brilliant Blue staining (Figure 1B, lanes P). The bands of the eight channels were clearly observed in the purified proteoliposome fraction, with a particularly high production of KCND2 and KCNH2. The mobility of the bands was roughly consistent with the expected molecular weight of each channel (KCNA1; 56 kDa, KCNA2; 40 kDa, KCNA5; 67 kDa, KCND2; 70 kDa, KCNH2; 127 kDa, KCNK2; 45 kDa, KCNK4; 43 kDa, KCNK5; 55 kDa).

The integration of the channels into the liposomes was confirmed by using density gradient centrifugation. After ultracentrifugation, proteoliposomes floated to the top of the iohexol gradient, while the soluble proteins and aggregates stayed at the bottom of the tube (Figure 1A). All of the channels were detected in the top fractions and not in the bottom (Figure 1B, lanes T and B), indicating that cell-free synthesized channels were associated with liposomes.

### 3.2. PLB Assay on Cell-Free Synthesized KCNK2

To demonstrate that our cell-free synthesized channels are functional, we measured the channel activity of the potassium ion channels using a PLB assay. A DPhPC lipid membrane was stretched over the sensor pore, and purified channel proteoliposomes were added to a measuring buffer in the chamber. Proteoliposomes spontaneously fuse with the lipid membrane over time. When the channel fused to the lipid membrane opens, ions travel through it, and a current signal is observed.

We selected KCNK2 (TREK and TREK-1), a well-studied K_2P_ channel, as our sample. It is known to open with changes in voltage and temperature [33,34]. After adding the cell-free synthesized KCNK2 proteoliposomes to the chamber of the PLB device and incubating them with a voltage applied, the current signals were observed. The conductance of the KCNK2 channel at 25 °C and 37 °C was measured by varying the voltage from 100 mV to −100 mV, respectively (Figure 2A,B). Under both temperature conditions, the frequency and current amplitude of the pulses changed with voltage. The I-V plots clearly showed the voltage dependency of KCNK2 (Figure 2C). When the temperature was increased from 25 °C to 37 °C, the average channel conductance increased from 93.1 pS to 112.4 pS. The conductance at 25 °C is consistent with the previously reported results of a mouse TREK-1a channel obtained using a single-cell patch clump [35].

### 3.3. Cell-Free Synthesis of 250 Human Channel Proteins

In order to attempt the cell-free synthesis of as many human channel proteins as possible, we comprehensively collected human channel genes from the cDNA resources of the Kazusa DNA Research Institute [24] and the MGC cDNA clone set [22], collecting 250 human channel cDNAs overall. The summary is presented in Figure 3A. The details of each individual channel are also provided in Appendix A. Potassium ion channel α-subunits were the most abundant, with 72 clones (28.8%); the cation channels (45 clones) and anion channels (29 clones) followed. Gap junction proteins and porins were also collected as they are membrane proteins involved in substance permeation in a broad sense. Of the 279 ion channels listed in the Guide to PHARMACOLOGY database (version 2022.2) [36], 218 channels, accounting for 78%, were included in the channels collected (Appendix A). Among the 330 ion channel genes listed in the HGNC database (version 2022-11-12) [37], our collections include 229 genes (69%). In addition to the subunits forming pores, 32 clones coding regulatory subunits are also included in the set.

These clones were cell-free synthesized as fusion proteins with a FLAG tag at their N-terminus. To efficiently prepare the transcription templates for a large number of channels, we amplified the linear DNA fragments using PCR from bacterial stocks harboring the respective expression plasmids. Considering the large number of channels to be synthesized, we synthesized these channels using the bilayer method (Figure 3B) in a 96-well plate format. A cell-free system can be easily parallelized and miniaturized, making it suitable for synthesizing a large number of samples. Liquid handling with a 96-well pipetting system also supports multi-sample synthesis experiments.

The synthesis of the channels was confirmed by using Western blotting to detect the FLAG tag (Figure 3C). Overall, 238 channels (95.2%) of the 250 channel proteins were successfully synthesized and detected at sizes close to those expected. The size of the KCNMA1 band was twice that of the expected size. The bands of five channels were detected as having smaller than expected sizes. No bands were detected in six clones (2.4%). We attempted to improve the synthesis of KCNB2, for which no band was detected, as seen in Figure 3C, by using codon optimization. The DNA of KCNB2 was replaced by the sequence with a codon set optimized for the human expression system and prepared using gene synthesis. The codon-optimized KCNB2 sequence (Appendix A) was inserted into the pEU-FLAG-GW plasmid and transcribed and translated. As a result, Western blotting revealed that KCNB2 channels with the correct size were synthesized (Appendix A).

### 3.4. Functional Evaluation of 47 Cell-Free Synthesized Voltage-Gated Potassium Ion Channels by PLB Assay

We then evaluated the activity of the synthesized channels by using a PLB assay. Since it is not possible to evaluate all of the channels using the same method, we focused on the voltage-gated potassium ion channels, the most abundant family among all of the channels. These channels can be evaluated by the same method by simply applying a voltage, which opens them even without a ligand. Our human channel collection contained 47 voltage-gated potassium ion channels (Appendix A). Electrically silent K_V_ subunits, such as KCNG1 (K_V_6.1) and KCNS1 (K_V_9.1), which are thought to show no channel activity by themselves, were excluded. A summary of the PLB measurement results is shown in Figure 4A and Appendix A, and the representative current signals for each channel are shown in Appendix A. We performed the PLB measurements twice with a planar lipid bilayer composed of DPhPC. The channels that did not show a current signal were measured two additional times with an asolectin membrane. If a current signal was observed in either measurement, the channel was considered active. We defined a current signal as normal if the current signal was continuously observed, the magnitude of the current signal was voltage-dependent, and the current stopped running when the voltage dropped (Figure 4B). A normal current signal was observed in 23 channels (48.9%). About 30% of the channels showed abnormal current signals, including partially open pores (12.8%, Figure 4C), temporarily open pores (8.5%, Figure 4D), not being voltage-dependent (4.3%, Figure 4E), and baseline shift over time (2.1%, Figure 4F). Including the irregular signals, approximately 80% of the potassium ion channels were opened by using voltage stimulation. The synthesis of the remaining nine channels was high enough but did not respond in four trials (Figure 3C).

### 3.5. Cell-Free Synthesized KCNB1 and KCNS3 Form a Heterocomplex

To demonstrate that the cell-free synthesized channels formed heterocomplexes on the liposome membranes, we co-synthesized KCNB1 and KCNS3 as a known heterocomplex combination [38,39]. Equal amounts of mRNA-encoding AGIA- and FLAG-tagged KCNB1 and one of FLAG-tagged KCNS3 were mixed and added into a cell-free translation mixture. The proteoliposomes were solubilized using *n*-dodecyl-β-D-maltoside and applied to immunoprecipitation with an anti-AGIA antibody and Protein G magnet bead. Western blotting detected both AGIA-FLAG-KCNB1 (150 kDa) and FLAG-KCNS3 (60 kDa) (Figure 5), indicating that the co-synthesized KCNB1 and KCNS3 formed a complex on the membrane and co-precipitated after solubilization. The signal of KCNB1 was slightly stronger than that of KCNS3, although exact quantification was not possible. When FLAG-KCNS3 was synthesized independently, it did not precipitate.

## 4. Discussion

This study demonstrated that a wide variety of human channel proteins could be synthesized using a wheat germ cell-free membrane protein synthesis system, not only simple and small proteins but also those of large sizes and complex structures. The Western blotting results evidenced that 95% of the 250 channels were synthesized successfully (Figure 3C). The channels with large molecular weight and numerous transmembrane helices such as ITPR3 (294.8 kDa, 6 TM), CACNA1G (262.5 kDa, 24 TM), SCN5A (222.7 kDa, 24 TM), and CACNA1S (212.3 kDa, 24 TM) were synthesized as polypeptide chains with approximately predicted sizes. The success rate and range of amount differences of cell-free synthesized channels in this study are comparable to the results of the previous report in which a huge number of human soluble proteins were synthesized using a wheat cell-free protein synthesis system [40]. On the other hand, a small number of channels were not synthesized in Figure 3C. The productivity of cell-free protein synthesis systems can be affected by template nucleic acid sequences and polypeptide products. For example, translation of human CLDN-5 protein is inhibited by the template mRNA sequence with extraordinarily high GC content [41]. Regulatory nascent peptides can arrest translation or modulate the translation rate [42]. Here, we demonstrated that the synthesis of KCNB2 was drastically improved by using the codon optimized template DNA (Figure 3C and Appendix A). This result indicates that it is the template nucleic acid sequence rather than the polypeptide chain sequence that is responsible for the poor translation of KCNB2. However, it is not clarified what factors prevent translation, since the GC content of the KCNB2 cDNA is 51%, and it does not have a distinctive repeat sequence.

The cell-free synthesis of channel proteins has several advantages, as proteins can be prepared outside of a cellular context. In a cell expression system, the overexpression of channels often affects intracellular homeostasis, resulting in low expression and poor cell proliferation. However, in a cell-free system, translation efficiency is not affected by the synthesized products. Another advantage of cell-free systems is that the lipid composition of the liposomes can be modified freely. By using various purified lipids or lipid mixtures instead of asolectin, a mixture of natural lipids, studies could be conducted to evaluate the effects of lipids on channel activation or to reconstitute an optimal membrane environment for the channel. On the other hand, there are several limitations that should be considered when using cell-free synthesis for channel protein preparation. First, the huge channels described above, even if they are successfully synthesized as polypeptides, need to be verified regarding whether they are properly folded and maintain functional structures on the liposome membranes. The second limitation is the integration of the channels into the liposome membrane. The results in Figure 3C are from the unpurified proteoliposomes, and it was not confirmed whether each individual channel associates with the liposome membrane. Association with scaffolds such as liposomes is important for the conformation and function of cell-free synthesized membrane proteins [7]. It was reported that membrane proteins with two or more transmembrane helices associate with the liposome membrane with a high probability in cell-free system with liposomes [8]. However, the association with the liposome membrane would need to be confirmed using density gradient (Figure 1A) in order to optimize the synthesis and functional expression of individual channels. Finally, in many channels, bands of different sizes were detected in addition to the expected size bands. Considering that the FLAG tag used for the detection of Western blotting is located at the N-terminus of the recombinant protein, it is possible that the frameshifts of the ribosome during translation resulted in truncations of different sizes.

The results of the PLB assay revealed that the majority of the cell-free synthesized 47 voltage-gated potassium ion channels formed channel complexes on the membranes and had channel activity (Figure 4A). Half of them exhibited reasonable current signals. Another 30% were observed to open in response to voltage changes, although the openings were incomplete; otherwise, the signals were not stable. Although there are still issues to be resolved, we believe that it is possible to synthesize functional channels using a cell-free synthetic system, as a voltage-dependent opening was detected in as many as 80% of the channels. In addition to potassium ion channels, it is worthwhile to attempt a functional analysis of other families of channels using cell-free synthesized recombinant channels.

Optimizing the sample preparation or measurement conditions may improve the probability of measurement success, stabilize the current signal, and allow for the correct evaluation of channel functions using a PLB assay. In this study, all of the channels were assayed under uniform conditions, which were not necessarily optimal for the individual channels. Optimization regarding the lipid composition of the planar lipid bilayer, the choice of buffer, and the lack of elements necessary for activation may improve the measurement of channels with unstable or undetectable signals. Other conditions related to cell-free synthesis also worth considering include the lipid composition of the liposomes, the amount of mRNA and liposomes in the translation reaction mixture, the translation temperature, and the translation time. The lipid composition of the liposomes seems to be a particularly important factor. The differences in the lipid composition of asolectin and human plasma membranes could lead to incorrect conformation and the inactivation of these channels. It is one of the advantages of cell-free systems that the composition of the lipid scaffold can be changed as desired. The use of various purified lipids or lipid mixtures instead of asolectin, which are mixtures of natural lipids, would allow for the reconstitution of an optimal membrane environment for the channel and an evaluation of the effects of the lipids on channel activation. In addition, it is not possible to determine from these results alone how many of the synthesized channel molecules possess the correct conformation and express channel activity. This is because PLB essentially measures the current generated by single-molecule channel complexes fused to a lipid membrane stretched over the electrode pore. Quantitative analysis of populations of channel molecules using cryo-electron microscopy and other techniques will help to resolve this issue. The orientation of the channels synthesized on liposomes must also be taken into consideration. Cell-free synthesized channels are presumed to be inserted into liposomes in a random orientation. Although orientation is not a major issue related to the PLB measurements of the voltage-gated potassium ion channels in Section 3.4, the PLB analysis of certain types of channels, such as inward rectifier potassium channels, should first confirm the orientation of the channels integrated into the planar lipid bilayer.

Due to advances in instrument miniaturization, simplicity, and multiple sample handling, PLB has recently become an easy-to-use technique. The preparation of good-quality recombinant ion channel samples is conducive to the wider use of PLB, which could be addressed by cell-free protein synthesis. One of the advantages of cell-free synthesis of channels is that regardless of their localization, whether they are plasma membrane channels or organelle membrane channels, they can be synthesized in the same way on artificial scaffolds. The combination of cell-free synthesized recombinant channels and PLB will greatly assist the functional analysis of channels that are hard to analyze using patch clamp, such as the channels localized on organelle membranes or channels that are extremely difficult to be expressed in cells.

The results of the KCNB1 and KCNS3 co-synthesis indicate the potential of the in vitro synthesis and utilization of heterocomplex channels. Most channels make homomeric complexes, but some channels may make heteromeric complexes under certain conditions [43,44]. For example, KCNB1 (K_V_2.1) normally forms a homotetramer expressing channel activity; however, it can also form heterotetramers with electrically silent KCNS3 (K_V_9.3) with a 3:1 stoichiometry [38,39]. One of the features of cell-free systems is that multiple mRNAs can be easily added to produce protein complexes [45]. It is theoretically possible to synthesize channel heterocomplexes in vitro by adding the mRNAs of the subunits into the translation reaction. However, it is unknown whether cell-free synthesized channel subunits are embedded in the same liposome and form a complex on those membranes. The results of the solubilization and immunoprecipitation of the co-synthesized KCNB1 and KCNS3 (Figure 5) strongly suggest that these channels formed a complex on the liposome membranes. In vitro co-synthesis is a useful method for comparing the function of various combinations of heteromeric channel complexes.

The Western blotting results of the 250 cell-free synthesized channels shown in Figure 3C are very informative for researchers who wish to scale up the production of specific channels using cell-free synthesis. However, it should be noted that the experiment was performed using a bilayer method (Figure 3B), which is suitable for the synthesis of multiple samples in small volumes. Higher yields can be achieved using the dialysis method or the bilayer dialysis method (Figure 1A) [14]. In addition, since cell-free systems have excellent scalability, it is easy to perform larger reactions or perform multiple reactions. If target channels can be easily synthesized in large quantities, a wide variety of applications are possible. One of the most promising applications is antibody development. Our previous studies have demonstrated that cell-free synthesized membrane proteins are effective immune antigens and that cell-free synthesized proteoliposomes can be injected into mice to develop highly specific or inhibitory antibodies [13,41]. We succeeded in developing inhibitory antibodies to ORAI1 (CRACM1) by immunizing mice with cell-free synthesized proteoliposomes [17]. Cell-free synthesized channels are a solution to one of the bottlenecks in the development of channel antibodies.

Finally, we wish to discuss the potential for a comprehensive collection of the human channel proteins developed in this study. The wheat cell-free protein synthesis system is highly reproducible and suitable for the simultaneous preparation of multiple recombinant proteins. We refer to such a set of recombinant proteins as a protein array. Protein arrays range from large-scale arrays that comprehensively cover the proteins of a certain species to relatively small-scale arrays that focus on protein families with specific functions or structural units. The set of 250 recombinant human channels is a relatively compact protein array; however, it may be an important tool for drug discovery because channels are important drug target proteins. Protein arrays are profoundly useful in comprehensive analysis, such as the discovery of enzymes or substrates, the identification of binding partners, antibody specificity analysis, and the target discovery of small-molecule drugs [41,46,47,48]. A promising application for the channel array is the discovery of drug seeds by affinity selection. However, liposomes may nonspecifically adsorb liposoluble drugs, so highly hydrophilic compounds should be targeted. It may also be effective in evaluating the specificity of antibody drug seeds targeting channels. In basic research, it could be applicable for a comprehensive analysis of the post-translational modifications of the channels.

## 5. Conclusions

The wheat cell-free system successfully synthesized 95% of the 250 human channel proteins. Approximately 80% of the 47 voltage-dependent potassium ion channels showed channel activity upon voltage change in the PLB assay. The activation of the cell-free synthesized channels by temperature change and heteromeric complex formation was also observed. In conclusion, these results indicate that the cell-free synthesized channels can provide a promising research tool. We expect that cell-free technologies can facilitate basic research and drug discovery studies targeting channels in a complementary manner than possible by traditional analytical methods using cells and patch clamps.

## Figures and Tables

**Figure 1 membranes-13-00048-f001:**
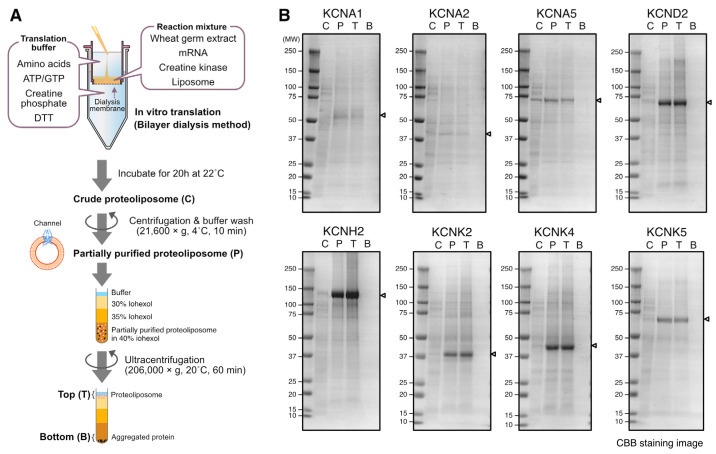
Interaction between cell-free synthesized channels and liposomes. (**A**) Scheme of cell-free synthesis and purification of channel proteoliposomes. (**B**). Proteoliposome samples subjected to density gradient centrifugation. Samples were subjected to SDS-PAGE with 5–20% gradient gels followed by Coomassie brilliant blue staining. C: crude translation mixture; P: partially purified proteoliposome; T: top fraction of density gradient; B: bottom fraction of density gradient. Arrowheads show the position of the band of each channel.

**Figure 2 membranes-13-00048-f002:**
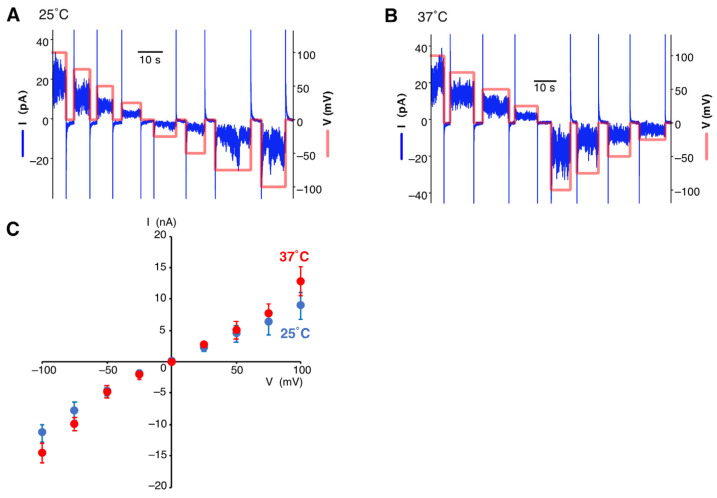
Single-channel current recording of cell-free synthesized KCNK2 channel at 25 °C (**A**) or at 37 °C (**B**). The blue line and main axis indicate current signal, and the red line and secondary axis show voltage. Bars indicate time scale. (**C**) I-V plots. The current record data at 25 °C (blue) and 37 °C (red) were analyzed by Clampfit 10.4 software. Results are shown as the mean ± S.E. (*n* = 6).

**Figure 3 membranes-13-00048-f003:**
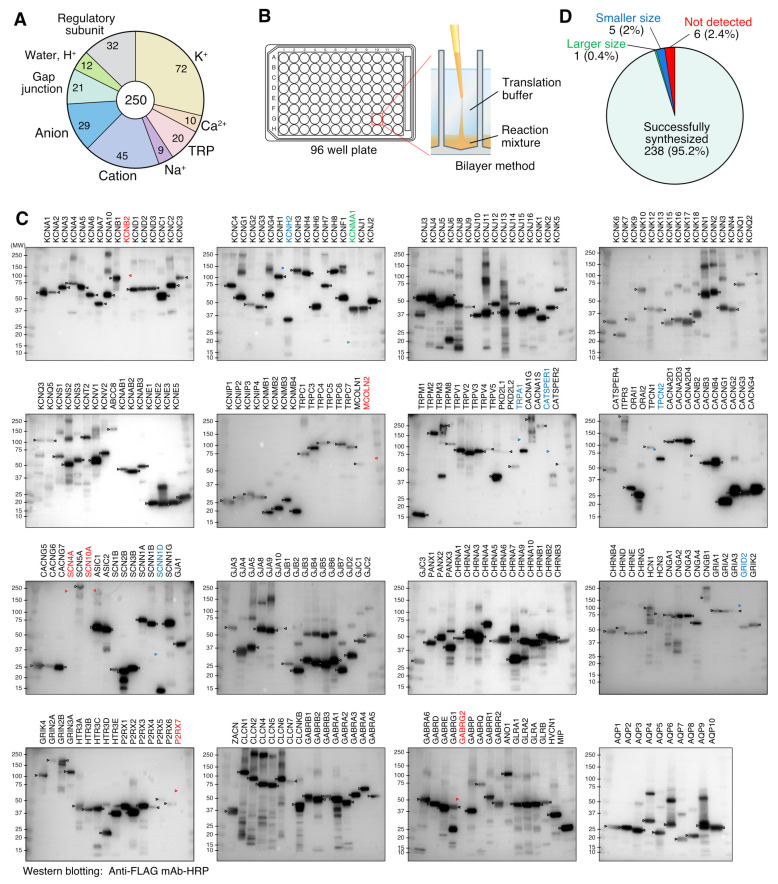
Cell-free synthesis of 250 human channels. (**A**) Composition of human channels synthesized in this study. (**B**) Illustration of bilayer method in 96-well plate. (**C**) Western blotting image of cell-free synthesized human channels. FLAG-tagged channels were detected by horseradish peroxidase-conjugated anti-FLAG antibody. Channels with names in black were well synthesized. Black arrowheads indicate bands of synthesized channels. Channels named in red have no polypeptide detected, where red arrowheads show the predicted mobility of the channels. Channels named in blue had bands of smaller size than expected (blue arrowheads). Channel in green had bands of larger size than expected (green arrowhead). (**D**) Summary of Western blotting result.

**Figure 4 membranes-13-00048-f004:**
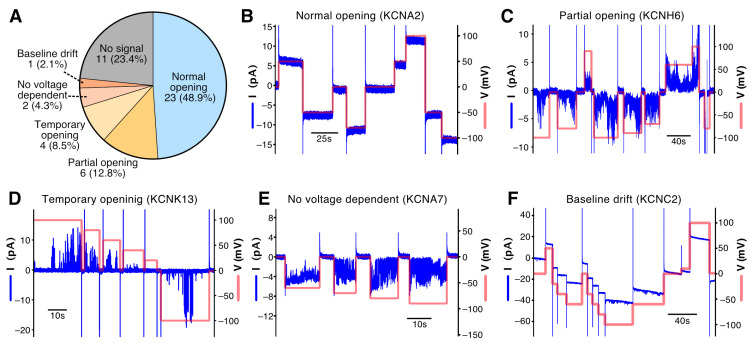
PLB assay of 47 cell-free synthesized voltage-gated K+ channels. (**A**) Summary of PLB assay. Representative current signals of normal opening (**B**), partial pore opening (**C**), temporary pore opening (**D**), no voltage-dependent opening (**E**), and baseline drift (**F**) are shown.

**Figure 5 membranes-13-00048-f005:**
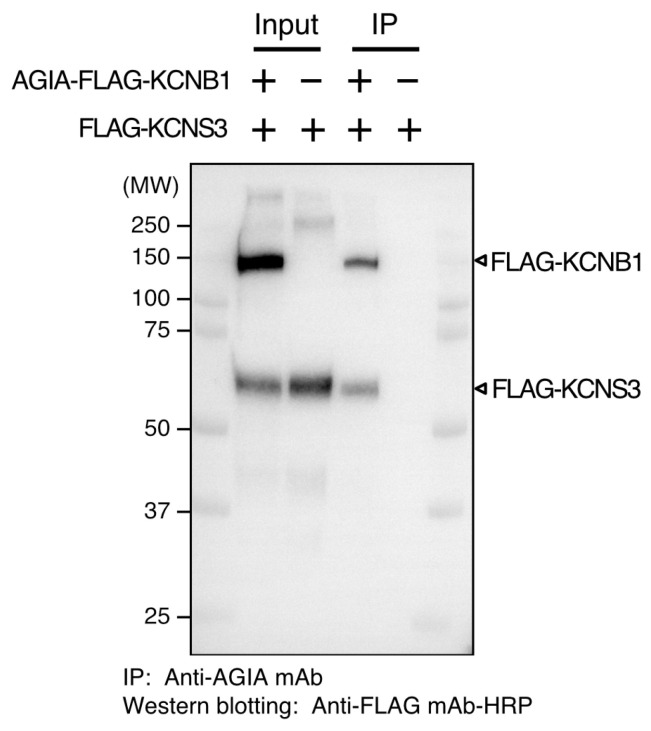
Production of heteromeric channel complex by co-synthesis. Co-synthesized AGIA-FLAG-KCNB1 and FLAG-KCNS3 proteoliposomes were solubilized and subjected to immunoprecipitation (IP). IP was conducted by anti-AGIA monoclonal antibody. Western blotting was detected using a horseradish peroxidase-conjugated anti-FLAG monoclonal antibody.

## Data Availability

Not applicable.

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
