# Peer review of "Evaluation of Cell-Free Synthesized Human Channel Proteins for In Vitro Channel Research"

_membranes, 2022, doi:10.3390/membranes13010048_

Round 1

Reviewer 1 Report

Overall, the authors provide a valuable study describing the expression of membrane channels using a liposome-supplemented cell-free protein synthesis system. While the work is of interest to the readership, there are a few major changes in wording of assertions in the text, methods, and results that should be completed prior to publication.

Major changes:

Assertions in Text:

Line 12 should revise “whether cell-free synthesis method can be applied universally to channel preparation” to “to which classes of membrane proteins cell-free synthesis can be applied.” No study could show “universal synthesis”, only what classes of molecules can be best made by a given method and how methods can be changed to improve synthesis. Line 56 same thing.

Methods:

Overall, the methods section is somewhat lacking in detail that would be required to recapitulate the work, which should be the aim of the work. Effort should be taken to provide as much information as is possible. For example, is there anything that can be disclosed about the reaction mixtures or components (such as concentrations), particularly that is relevant to their ability to produce the proteins of interest?

Results:

Overall in the results, the authors should attempt to compare and contrast channels that successfully expressed and those that didn’t. For example, why did KCNA2, KCNA1 expressed poorly while other protein in figure 1 expressed fairly well? Why were some channels functional in Table S2? Why were no bands detected for 6 of the clones? 

Figure 2 and associated text: Can the authors provide an estimate of the functional titer of the material either based on a standard made in living cells or based on reference values? Is 100% of the isolated material active or a limited portion?

Figure 3: This is already an impressive and large experiment, however, it should really have been performed analyzing both total and soluble fractions of the cell-free reactions with and without addition of the liposomes. The authors should not just be showing expression, but that the addition of liposomes enables the production of soluble protein. The assertion that the liposomes are important for the correct expression of these proteins seems important to the paper in its current form, so this experiment should be revised to include this control to support that claim.

Section 3.4: This is an impressive dataset, the authors should be recognized for the hard work it took to generate this data. Again, however, the authors should comment on why certain channels showed activity while others did not.

Supplement: 

The authors did a great job of disclosing the peptide sequence for each protein. They should also provide the full sequence of the expression template encoding each cDNA. It’s now widely recognized that full sequences, not just primers should be provided as it allows other members of the scientific community to replicate and build upon the work. 

Minor changes:

Line 62. “quality” is not very descriptive so sounds like the authors will just be restating their assertion that cell-free synthesis is useful for channel research. Should be discussing specifically what applications are most promising based on these findings and, most importantly the limitations of cell-free synthesis of channels and if found, methods to overcome them. 

Line 78: authors should describe the source of and methods to prepare azolectin liposome. 

Line 83, can the authors describe the reason for layering in the dialysis cup rather than mixing. Is there a reason for 22C? 

Line 91: authors should describe what is meant by “purified proteoliposome suspension”

Line 187: please write out CBB staining abbreviation, this is not very commonly used. 

Figure 1: centrifugation should always be reported in xg rather than rpm. 

Line 237: Can the authors mention why these channels (218 channels out of the 279 in PHARMACOLOGY and 229 out of 330 in HGNC) were selected?

Line 246: The authors should state not that linear DNA fragments from PCR starting from bacterial stocks can be used, but that it was used in this study. 

Line 254: Can the authors comment on what the larger and smaller bands are likely to be? Truncations or protease cleavage or oligomers? What patterns can be seen in this data? What classes of molecules express best?

Line 334: How would the authors propose to optimize the expression conditions?

Line 349: addresses -> addressed

Line 378: antibody production should be changed to antibody development or discovery

Citations:

Recommend citing additional works on protein lipid nanodiscs, a commonly described alternative to the liposomes used in this study. A wide variety of works by Sligar and colleagues describe these nanodiscs https://febs.onlinelibrary.wiley.com/doi/full/10.1016/j.febslet.2009.10.024 and additional works show the use of these nanodiscs in cell-free membrane protein synthesis. Other works can be cited, but this is one example: https://onlinelibrary.wiley.com/doi/epdf/10.1002/bit.26502

Reviewer 2 Report

The key concept of this manuscript is that the authors successfully synthesized human channels using their wheat cell-free membrane protein production system, and evaluated their functions in liposomes. Purifying functional membrane proteins such as channels is generally difficult, thus this study will provide useful tools to promote membrane protein studies. I believe the following specific comments will help the authors improve their work.

Major comments

1. Is the channel orientation properly controlled in the proteoliposome? Please mention this point in the discussion.

2. Please specify the lipid composition for the proteoliposome in the materials and methods. The author describes in the discussion that ‘One of the advantages of cell-free synthesis of channels is that regardless 350 of their localization, whether they are plasma membrane channels or organelle membrane 351 channels, they can be synthesized in the same way on artificial scaffolds. (Line 350-352)’ Please also discuss whether it is possible to apply their technique for different cell/organelle membranes with different lipid compositions, or discuss a technical limitation.

Minor comments

Please carefully go through the English language once more (definite article, singular/plural, etc.)

Line 86: with an HBS

Line 127: as a template

Line 160: to a new tube

Line 164: gentle rotation

Line 167: gentle rotation

Line 228: Results are shown as

Line 231: as much channel protein 

Line 243: Considering a large number of channels 

Line 349: which could be addressed by

Round 2

Reviewer 1 Report

Overall, the authors responded well to all comments and this work is ready to accept in principle with very minor revisions.

Building off my previous comment on universality, Line 73 should be revised to "reported cases are still insufficient to determine which channel protein proteins classes to which cell-free synthesis can be applied." Again, I do not believe the word universial should be used here. 

I do still believe that the study would have benefited from additional description of methods and the authors hypotheses on why some proteins express differently from others if the authors are able to add them. I also feel that additional experiments to determine if the solubility of proteins expressed in Figure 3 was dependent on the addition of liposomes to increase the impact of the work, but this would require significant additional experiments that need not delay publication of this work.
